# New Limits for Stability of Supercapacitor Electrode Material Based on Graphene Derivative

**DOI:** 10.3390/nano10091731

**Published:** 2020-08-31

**Authors:** Veronika Šedajová, Petr Jakubec, Aristides Bakandritsos, Václav Ranc, Michal Otyepka

**Affiliations:** 1Regional Centre of Advanced Technologies and Materials, Faculty of Science, Palacký University, Šlechtitelů 27, 78371 Olomouc, Czech Republic; veronika.sedajova@upol.cz (V.Š.); a.bakandritsos@upol.cz (A.B.); vaclav.ranc@upol.cz (V.R.); 2Department of Physical Chemistry, Faculty of Science, Palacký University, 17. Listopadu 1192/12, 77900 Olomouc, Czech Republic

**Keywords:** graphene acid, supercapacitor, pseudocapacitance, cycling stability

## Abstract

Supercapacitors offer a promising alternative to batteries, especially due to their excellent power density and fast charging rate capability. However, the cycling stability and material synthesis reproducibility need to be significantly improved to enhance the reliability and durability of supercapacitors in practical applications. Graphene acid (GA) is a conductive graphene derivative dispersible in water that can be prepared on a large scale from fluorographene. Here, we report a synthesis protocol with high reproducibility for preparing GA. The charging/discharging rate stability and cycling stability of GA were tested in a two-electrode cell with a sulfuric acid electrolyte. The rate stability test revealed that GA could be repeatedly measured at current densities ranging from 1 to 20 A g^−1^ without any capacitance loss. The cycling stability experiment showed that even after 60,000 cycles, the material kept 95.3% of its specific capacitance at a high current density of 3 A g^−1^. The findings suggested that covalent graphene derivatives are lightweight electrode materials suitable for developing supercapacitors with extremely high durability.

## 1. Introduction

There is tremendous interest in developing different kinds of power supplies owing to the escalating consumption of energy and the need for (mobile) energy storage [1]. Energy devices with high power and energy densities are increasingly important in fields such as industry, the automotive sector and energy supplies, e.g., for portable and medical devices [2]. Supercapacitors (also known as ultracapacitors) offer a higher power density than batteries and a higher energy density than conventional capacitors [3]. These attributes, together with high stability and rather low costs, make them suitable candidates for energy storage devices. Based on their charge storage mechanism, they can be divided into two main classes: electrochemical double-layer capacitors (EDLCs), where the main contribution to the overall capacitance comes from the charge accumulated at the electrode/electrolyte interface [4], and pseudocapacitors, where the charge storage mechanism is driven by fast and reversible faradaic (redox) reactions [5]. Pseudocapacitive materials with a typical faradaic response are able to store more energy (per mole of material) than EDLCs since redox processes occur both at the surface and in the interior of the electrodes. The capacitance of pseudocapacitors can be up to 100 times higher than that of EDLCs [6]. Since the intrinsic properties of redox-active materials can significantly influence both capacitance and rate performance, the selection of a suitable material is of paramount importance.

Common materials that undergo faradaic responses include conventional transition metal oxides, such as RuO_2_ [7,8], MnO_2_ [9,10,11,12] and Co_3_O_4_ [13,14] and various conducting polymers [15,16,17,18]. However, these compounds often fail to meet the necessary criteria for practical applications, such as low costs and stability during life cycling [19]. By contrast, carbon-based materials offer the advantages of reasonable prices and a suitable capacitive response. Moreover, the surface of carbon derivatives can be easily modified by different functional groups [20,21]. Such modifications can boost the electrochemical stability as well as capacitive performance. For example, porous carbon has been modified with nitrogen- [22,23,24], oxygen- [25,26,27] and phosphorus-containing functional groups [28,29]. As oxygen-containing groups often occur in carbon materials [30], their use may offer a promising strategy for improving the capacitive performance. The insertion of oxygen-containing groups can have positive effects on a material’s wettability and pseudocapacitive behavior. In particular, the wettability of the material by an electrolyte increases, and the pore volume becomes more accessible to electrolytes. Therefore, the electrostatic interactions between the electrode and electrolyte ions spread over a larger surface, enhancing the capacitance [21]. Oxygen-containing functionalities may also be involved in fast redox reactions, which can enhance the overall capacitance by the pseudocapacitance contribution [31]. However, it should be noted that some disadvantages counteract the benefits of introducing oxygen-containing groups. The most commonly occurring shortcomings include potential organic electrolyte decomposition, high self-discharge rates, increased leakage current and decreased conductivity [32].

Attention has been paid to optimizing carbon-based materials and improving their lifetime stability. For instance, Cao et al. studied graphene derivatives prepared at different temperatures containing different amounts of oxygen-containing functional groups. The best sample displayed a life cycle stability of 10,000 cycles, with capacitance retention of 98% [33]. Cherusseri et al. showed that oxygen-containing functional groups in hierarchically mesoporous carbon nanopetal based electrodes significantly improved their lifetime stability, reaching 28,900 cycles [34]. Chen et al. investigated the influence of oxygen-containing groups on the supercapacitive performance of nitrogen-doped graphene. Their derivatives exhibited good lifetime stability over 5000 cycles, with capacitive retention of 86.36% and 91.14% [35]. The importance of oxygen-containing functional groups was also shown for graphene modified with triethanolamine by Song et al. The lifetime cyclic stability reached 10,000 cycles, with capacitance retention of 91.7% [36]. However, since supercapacitor devices need to withstand a huge number of charge/discharge cycles, there is scope for further improvements.

In the present study, we investigated the pseudocapacitive behavior of graphene acid (GA) [37] as a potential supercapacitor electrode material with a particular focus on synthesis reproducibility, i.e., batch-to-batch variability and life cycle stability. GA is a conductive, hydrophilic graphene derivative that is well-endowed with oxygen-containing carboxyl groups homogeneously distributed over the graphene surface [37]. GA can be used as a promising material in electrochemical sensing [38], catalysis [39,40,41] and electrocatalysis [42]. Its properties as an electrode supercapacitor material were tested in a neutral salt environment together with cyanographene [38]. Here, we provide a significantly more interesting look at a pseudocapacitive behavior of GA obtained in 1 mol L^−1^ sulfuric acid. A comparison of the properties of three independent GA batches showed the high reproducibility of its synthesis. We also demonstrated that it had very large specific capacitance retention after 60,000 cycles (95.3%) recorded at a high current density of 3 A g^−1^ in a two-electrode cell, suggesting that electrodes based on GA may offer outstanding rate stability. These findings place covalent graphene derivatives bearing oxygen-containing functional groups at the forefront for achieving high-durability eco-friendly electrode materials (without the presence of any heavy-metal elements) that can be used in supercapacitors.

## 2. Materials and Methods

### 2.1. Reagents and Materials

Graphite fluoride (>61 wt % F), sodium cyanide (p.a. ≥ 97%), *N,N*-dimethylformamide (≥98%), sodium sulfate (p.a. ≥ 99%) and potassium hydroxide (BioXtra ≥85%) were purchased from Sigma-Aldrich (St. Louis, MO, USA). Nitric acid (Analpure 65%) and sulfuric acid (p.a. 96%) were obtained from Lach-Ner. Acetone (pure) and ethanol (absolute) were purchased from Penta, Czech Republic. All chemicals were used as delivered without any further purification. Ultrapure water (18 MΩ cm^−1^) was used for the preparation of all aqueous electrolyte solutions.

### 2.2. Synthesis of Graphene Acid

GA was synthesized from cyanographene similarly to a previously published method [37]. Briefly, graphite fluoride (120 mg) was dispersed in 15 mL *N,N*-dimethylformamide (DMF) and stirred for 2 days under a nitrogen atmosphere. Next, the mixture was sonicated for 4 h and then left under stirring overnight. Afterwards, 800 mg of NaCN was added, and the mixture was heated at 130 °C for 72 h with a condenser and under stirring with a Teflon coated magnetic bar (400 rpm). At the end of the reaction, the reaction flask was left to cool down, and then washing steps were performed using a variety of solvents covering an extensive polarity window (DMF, acetone, ethanol, hot water, cool water and acidified water—3% solution of HCl). Materials were separated from the solvents using centrifugation (centrifuge Sigma 4-16K, Sigma Laborzentrifugen GmbH, Osterode am Harz, Germany) at 10,000 RCF. The final (pure) cyanographene was subjected to hydrolysis to obtain GA [37]. Concentrated nitric acid (65%) was added to a suspension of cyanographene in water under stirring in a ratio to obtain a 25% concentration. The mixture was heated at 100 °C for 24 h with a condenser and under stirring with a Teflon coated magnetic bar (400 rpm). The final product was left to cool down, and washing steps with cold water, hot water and acidified water were performed until precipitation of the product ceased. GA was separated from the solvents using centrifugation at 20,000 RCF.

### 2.3. Characterization Techniques

X-ray diffraction (XRD) measurements were recorded using X’Pert PRO MPD diffractometer (PANalytical) in the Bragg–Brentano geometry. A Co X-ray tube (iron filtered Co Kα radiation with λ = 0.178901 nm), fast X’Celerator detector, and programmable divergence and diffracted beam antiscatter slits were equipped. Fourier transform infrared (FTIR) spectra were measured on an iS5 FTIR spectrometer (Thermo Scientific Nicolet) using the Smart Orbit ZnSe ATR accessory. A drop of an ethanolic dispersion of the material was placed on a ZnSe crystal and left to dry to form a film at ambient temperature. Spectra were recorded by summing 50 scans acquired while nitrogen gas flowed through the ATR accessory. Baseline correction was used on the collected spectra. Raman spectra were recorded on a DXR Raman microscope using a diode laser with a 633 nm excitation line. X-ray photoelectron spectroscopy (XPS) was performed on a PHI VersaProbe II (Physical Electronics) spectrometer using an Al Kα source (15 kV, 50 W). The MultiPak (ULVAC–PHI Inc., Kanagawa, Japan) software package was used for evaluation of the obtained data. High-resolution transmission electron microscopy (HRTEM) images including STEM-HAADF (scanning transmission electron microscopy-high-angle annular dark-field imaging) analyses for elemental mapping of the products were acquired with a FEI Titan HRTEM microscope using an operating voltage of 200 kV. For these analyses, a droplet of a dispersion of the material in ultrapure H_2_O with a concentration of 0.1 mg mL^−1^ was deposited onto a carbon-coated copper grid and dried. Atomic force microscopy (AFM) images were obtained on an NTEGRA Spectra instrument (NT-MDT, Moscow, Russia) in tapping mode using NSG30 probes. In total, 5 µL of an ethanolic dispersion (*c* = 1 mg/L) of the analyzed nanomaterial was deposited on a SiO_2_ wafer and left to dry for 30 min, and then the sample was immediately scanned by AFM. Pore size and surface area analyses (Brunauer, Emmett and Teller; BET) were carried out using a volumetric gas adsorption analyzer (3Flex, Micromeritics, Norcross, USA) at −196 °C and up to 0.9626 bar. The sample was degassed under high vacuum conditions for 24 h at room temperature using high-purity gases (N_2_ and He). Conductivity measurements were performed using an Ossila four-point probe system operating over a current range of ±10 nA to ±150 mA (voltage range ±100 μV to ±10 V). The parameters used for the obtained results were as follows: range max 2000 μA; 8192 points per sample; target current was set to 0.5 mA, with a maximum voltage of 1 V with 0.1 V increments; 50 repeats were measured. A fluorine-doped tin oxide (FTO) substrate (1.5 cm × 2.5 cm) was modified with the sample by drop-casting, i.e., a 150 µL drop of a powder suspension (5 g L^−1^) was coated onto the substrate in the center and dried using an infrared light lamp (Helago, Parchim, Germany) to form a thin film, with a diameter of 11.8 mm and a thickness of 4 μm. During the measurement, the four probes were placed carefully on top of the dried film, centered and pressed slightly in order not to touch the FTO substrate.

### 2.4. Electrochemical Measurements

A MetrohmAutolab PGSTAT128N instrument (MetrohmAutolab B.V., Netherlands) controlled with the NOVA software package (version 1.11.2) was used for complex characterization of GA in a three-electrode setup. A BCS-810 battery tester (BioLogic Company, Seyssinet-Pariset, France) controlled with BT-Lab software (version 1.63) was used for two-electrode cell experiments. Sulfuric acid (*c* = 1 mol L^−1^) was used as a supporting electrolyte unless otherwise stated. All experiments were carried out at room temperature (22 ± 2 °C).

The three-electrode system consisted of a glassy carbon electrode (GCE) serving as the working electrode, a platinum wire electrode as a counter electrode and an Ag/AgCl (3 M KCl) electrode as a reference electrode. Electrochemical impedance spectroscopy (EIS) spectra were recorded using a 5 mV amplitude. All EIS spectra were recorded over the frequency range of 0.1 Hz to 10 kHz at open circuit potential (OCP). The GCE was modified as follows: a 10 µL drop of a powder suspension (2 mg mL^−1^) was coated onto the surface of the GCE electrode and allowed to dry at ambient temperature to form a thin film.

For the two-electrode system, a symmetrical supercapacitor device was constructed according to the following protocol. Briefly, the active material was homogeneously dispersed in ultrapure water (∼5 mg mL^−1^) and sonicated for 1 h. Next, 300 μL of the dispersion was drop-coated onto the surface of a gold disk electrode (diameter 18 mm) and dried under an infrared light lamp in a desiccator to achieve a mass loading of at least 1.2 mg cm^−2^. For assembly of the supercapacitor device, two gold disk electrodes with the same loading of active material were placed in an insulator sleeve (EL-Cell insulator sleeve, EL-Cell GmbH, Hamburg, Germany) using Whatman glass microfiber filter paper with a thickness of 0.26 mm as a separator. The separator membrane was soaked with 100 μL of electrolyte. Stainless steel plungers were used to press the electrodes, and the whole device was tightened and connected.

In the three-electrode setup, the specific capacitance of the active material (*C*_s_, F g ^−1^) was calculated from the galvanostatic charging/discharging (GCD) curves as follows (Equation (1)):(1)$$C_{s}=\frac{2i_{m}}{{\left(V_{f}-V_{i}\right)}^{2}}\int_{V_{i}}^{V_{f}}Vdt$$
where *i_m_* represents the current density in A g^−1^, “Vdt” is the integrated area under the discharge curves and *V*_f_ and *V*_i_ are the final and initial values of the potential range (V), respectively.

In the two-electrode setup, the specific capacitance of the cell (*C*_s_, F g ^−1^) was calculated using Equation (2):(2)$$C_{s}=\frac{I\times \Delta t}{m\times \Delta V}$$
where *C_s_* is the gravimetric capacitance of the electrode (F g^−1^), *I* is the discharge current (A), Δ*t* is the discharge time (s), Δ*V* is the potential window and *m* is the total mass of both electrodes (g).

## 3. Results and Discussion

The Bragg diffraction of GA displayed two broad peaks centered at 29° and 50° corresponding to the diffraction of the (0 0 2) and (1 0 1) planes (Figure 1a). The intense broad peak recorded at 29° related to a *d*-space value of 0.3579 nm, which was slightly higher than the equivalent value for the graphite structure (inset of Figure 1a). Such peaks suggest an amorphous structure containing stacked sheets with short-range order [43]. A Raman spectrum of GA revealed two peaks in the range of 1000 to 1800 cm^−1^. The D-band at 1336 cm^−1^ originated from a breathing mode of κ-point phonons of A_1g_ symmetry, whereas the G-band at 1605 cm^−1^ originated from the first-order scattering of E_2g_ phonons by sp^2^ carbon domains. The higher intensity of the D-band (*I*_D_/*I*_G_ ratio of 1.69) indicated atomic-scale defects, such as vacancies and grain boundaries [44,45], which commonly appear in highly functionalized graphene derivatives [46,47,48]. An XPS survey spectrum (Figure 1c) showed the dominant presence of C (76.7 at. %) and O (18.0 at. %), followed by N (3.5 at. %) and F (1.8 at. %) elements. The fluorine content probably originated from the graphite fluoride precursor, whereas the nitrogen content may have originated from unreacted nitrile groups, as cyanographene is a reaction intermediate of GA [37]. The deconvoluted C 1s XPS spectrum of GA (Figure 1d) exhibited six symmetric peaks corresponding to carbon atoms in different functional groups. The first two peaks located at 284.75 eV and 286.75 eV were assigned to sp^2^ C=C and sp^3^ C-C bonds, respectively, whereas the peaks located at 287.80, 288.84, 286.99 and 290.03 eV were attributed to carbon involved in C-O, O=C-O, C-N and C-F covalent bonds, respectively. The deconvoluted O 1s XPS spectrum (Figure 1e) exhibited two symmetrical peaks located at 531.62 and 533.32 eV, which were attributed to oxygen in C=O and C-OH covalent bonds, respectively. Therefore, the XPS analysis confirmed that GA contained a high content of oxygen-containing carboxyl functional groups, as supported by FTIR measurements (Appendix A) and in agreement with the literature [37]. BET measurements revealed the specific surface area of 18.6 m^2^ g^−1^ (Appendix A) with an average pore size of approximately 6.2 nm in diameter. It should be noted that the specific surface area in the dry state was low due to the tendency for restacking of individual sheets of graphene functionalized by hydrophilic groups [37,49].

The structural properties and composition of GA were further evaluated by microscopic methods. HRTEM indicated the layered nature of GA, with thin flakes of micrometer-scale lateral size (Figure 2a). Selected area electron diffraction (SAED) analysis displayed two well-developed rings, confirming the graphitic nature of GA with the presence of the (0 0 2) and (1 0 1) planes (Figure 2b). HRTEM imaging was also employed to provide more detailed information about the structure and topography of the chemical element composition (Figure 2c–e). Dark-field HRTEM images were used for energy dispersive X-ray spectroscopy (EDS) chemical mapping of carbon, oxygen, fluorine and nitrogen (Figure 2d,e), which all showed a homogeneous distribution over the surface, confirming the homogeneous surface functionalization of graphene. The energy-dispersive X-ray (EDS) spectrum of GA acquired during HRTEM analysis (Figure 2f) showed dominant peaks for C and O and small peaks for N, F, Na and Ca. The last two elements probably originated from metal ion impurities present in the water used for the sample preparation, as XPS did not detect these metal elements. An AFM image of a GA flake and the corresponding height profile are shown in Appendix A, respectively. A structural model of GA is shown Figure 2g. All the obtained data corroborated that the GA sample could be considered a layered system with a homogeneous distribution of all tested elements over the surface. The findings confirmed that GA is a graphene derivative homogenously functionalized by oxygen-containing carboxyl groups, in agreement with previous reports [37].

The reproducibility of chemical syntheses is important for practical applications, especially in nanomaterial chemistry. Thus, we prepared and tested three independent batches of GA. Both Raman and XPS spectroscopy confirmed minimal changes in chemical composition among the different batches (Appendix A). The high reproducibility of GA synthesis was also corroborated by the results of electrochemical experiments, as described below.

### Electrochemical Performance of Graphene Acid

A three-electrode setup including a reference electrode is able to describe the electrochemical behavior of a tested material more properly than a two-electrode cell can [52,53]. However, a two-electrode setup is closer to the operational conditions of real capacitor devices. Thus, we decided to use a three-electrode setup first to obtain an insight into the electrode/electrolyte behavior of the tested GA. Figure 3a shows a cyclic voltammetry (CV) response of GA in the presence of three different electrolytes over the potential range of 0.0 to 1.0 V at a constant scan rate of 20 mV s^−1^. Sulfuric acid has the ability to boost the current (capacitive) performance compared to a neutral or alkaline electrolyte of the same ionic strength [54]. The recorded CV scan of GA exhibited distortion of the “ideal” rectangular shape, with a pair of redox peaks located at 0.4–0.45 V. The slight potential separation (<0.1 V) between the anodic and cathodic peak potentials suggests that the electrochemical response of GA could be considered a highly reversible system accompanied by minimal or no structural changes. Such behavior is expected for a system containing oxygen-containing functional groups, as shown above for GA. Oxygen-containing functional groups could enhance capacitive performance; they increased the interaction of the tested material with the electrolyte and made the surface more accessible for ion-exchange reactions. Moreover, fast redox reactions of carbon surface functionalities may introduce an extra capacitance (pseudocapacitive effect) and boost the overall performance [31]. This pseudocapacitive effect was apparent in the CV and could be attributed to the presence of oxygen-containing groups (FTIR spectra of GA—Appendix A). The absence of redox peaks in the neutral or base-derived solutions could be attributed to the lack of protons in these electrolytes (the absence of H^+^ prevents the undergoing of the redox reaction of oxygen-containing functional groups). As the sweep rate was increased (Figure 3b), both the anodic and cathodic peak currents exhibited a significant shift, reflecting the ohmic contribution. During a linear scan of a constant rate, the current can be expressed according to a power law as follows (Equation (3)):(3)$$i=av^{b}$$
where *i* is the current, *a* and *b* are adjustable values and *v* is the scan rate. Thus, the *b* factor can be obtained from the slope of a linear fit when log *i* is plotted against log *v* at a fixed potential. A value of *b* = 0.5 indicates the current is controlled by semi-infinite diffusion processes, whereas *b* = 1 indicates that the current is surface-controlled (influenced by capacitive or nondiffusive processes) [55,56]. As shown in Figure 3c, the *b* factor was close to 1 (data for both the anodic and cathodic current peaks were collected at a constant potential of 0.4 V), suggesting strong surface-associated processes. The ratio between the surface-associated processes (e.g., redox reactions) and diffusion-controlled processes (e.g., EDLC contribution) was further evaluated using Trasatti’s method [57] (a detailed methodology description can be found in the Appendix A). As expected, the presence of oxygen-containing functional groups on the surface of GA reflected the high portion of the pseudocapacitive mechanism (55.9%). Since GA has the structure of layered graphene, the contribution from the EDLC mechanism (44.1%) was apparent as well. Figure 3d shows GCD curves recorded at current densities ranging from 0.5 to 10 A g^−1^. Their symmetric quasitriangular shape suggests the presence of redox-active material as well as a satisfactory capacitive response of GA, even at high current densities. In addition, three samples of GA were synthesized and tested to investigate the reproducibility of the synthesis (Figure 3e). The standard deviation of the results was found to be smaller than 10% (current densities ranging from 0.5 to 5 A g^−1^), suggesting the practical applicability of the GA sample. A small deflection was observed at the highest current density of 10 A g^−1^, for which the standard deviation was found to be 13%. A maximum specific capacitance (*C*_sp_) of 111 F g^−1^ (average for the three independent syntheses of the GA sample (S.D. 5.3%)) was obtained at a current density of 0.5 A g^−1^. The obtained results show that GA is a very competitive material comparable to previously published reports (Appendix A). EIS was also employed to evaluate the GA sample based on a modified Frumkin–Melik–Gaykazyan circuit (inset of Figure 3i). The suitability of this circuit for analyzing the EIS spectra was reflected in the good error of fit (Figure 3i). One can see from the magnified region of the Nyquist plot (Figure 3g) that in the high-frequency region, the EIS spectrum of GA did not exhibit a significant semicircle, indicating a small value of charge transfer resistance, *R*_ct_. According to the data fit, the *R*_ct_ was around 1.06 Ω. Such findings predict good conductivity for the GA sample. We also employed the four-point probe method to test the conductivity of GA. The obtained data confirmed a high conductivity for GA at 8.174 × 10^3^ Sm^−1^ and a rather low value of resistivity at 1.229 × 10^−4^ Ω m (sheet resistance 30.74 Ω square^−1^). An almost vertical line in the low-frequency region of the Nyquist plot (Figure 3f) indicated the nearly ideal capacitive response of GA, which could be attributed to fast ion diffusion [58]. The slight deflection from the parallel line was assigned to redox reactions of the oxygen-containing functional groups [5]. A Bode representation for a GCE modified with GA is depicted in Figure 3h. As can be seen, when the frequency decreased, the phase angle reflected a change from a resistive to a fully capacitive response in the mHz region (phase angle: ∼−84°). It should be noted that a phase angle close to −90° indicates an almost ideal capacitor.

To further assess the feasibility of GA for actual applications, a symmetric capacitor using 1 mol L^−1^ H_2_SO_4_ as the electrolyte was assembled. To avoid any capacitance interferences or boosting effects originating from binders or different additives, we used an aqueous suspension of GA. Figure 4a shows the GCD response of GA in the two-electrode configuration recorded at current densities ranging from 1 to 20 A g^−1^. As expected, all the GCD curves were more symmetrical compared to those of the three-electrode setup, suggesting the reduced influence of ion diffusion as well as satisfactory capacitive behavior. Faster ion diffusion is also demonstrated in Figure 4b, which shows a slow decline in the specific capacitance response across a wide range of current densities. Based on these observations, we also performed a rate stability test (Figure 4c) to evaluate the capacitance stability of the GA system. After 10 repeated GCD tests over a range of current densities from 1 to 20 A g^−1^, the capacitive response remained stable. Hence, it was possible to select a relatively high current density (3 A g^−1^) and cycle the system during a long-term experiment. The results confirmed the remarkable stability of GA after 60,000 cycles, with specific capacitance retention of 95.3% (Figure 4d). In practical applications, a single supercapacitor is not able to meet all the energy and power requirements. Therefore, two cells were connected in series (to boost the voltage range) or in parallel (to increase the energy storage density), as illustrated in Figure 4e. The GCD test recorded at 2 A g^−1^ revealed that the voltage window could easily be increased up to 2 V for two cells connected in series compared to a single supercapacitor operating at 1 V. On the other hand, connecting supercapacitors in parallel could enhance the energy storage density twice as well. To demonstrate that GA can be used as a candidate for supercapacitor applications, two cells were connected in series and used to power a red light-emitting diode. After charging the system to 2 V, the LED was illuminated without any obvious fading effects, as shown in Figure 4f (Figure 4f (left) shows the open circuit without the presence of GA-based supercapacitors, whereas Figure 4f (right) illustrates the closed circuit with fully charged GA-based supercapacitors).

## 4. Conclusions

GA is a graphene derivative prepared from fluorographene that offers a promising stable electrode material for supercapacitors. GA is well-equipped with oxygen-containing groups and exhibited an encouraging pseudocapacitive response in 1 mol L^−1^ sulfuric acid. By prepared three independent GA samples, we showed the high reproducibility of its synthesis and electrochemical performance. Owing to the unique features of its surface chemistry, GA exhibited good performance as a pseudocapacitive material, having an enormous rate and lifetime stability. GA could be cycled at least 10 times at various current densities without any capacitance loss. Such an attribute was accompanied by an enormous specific capacitance retention after 60,000 cycles (95.3%) recorded at a reasonably high current density of 3 A g^−1^ in a two-electrode cell system. The obtained results demonstrate that GA is a suitable candidate for preparing supercapacitors with exceptional lifetime stability. Generally, covalent graphene derivatives with selected functional groups should be considered a potentially important class of supercapacitor electrode materials.

## Figures and Tables

**Figure 1 nanomaterials-10-01731-f001:**
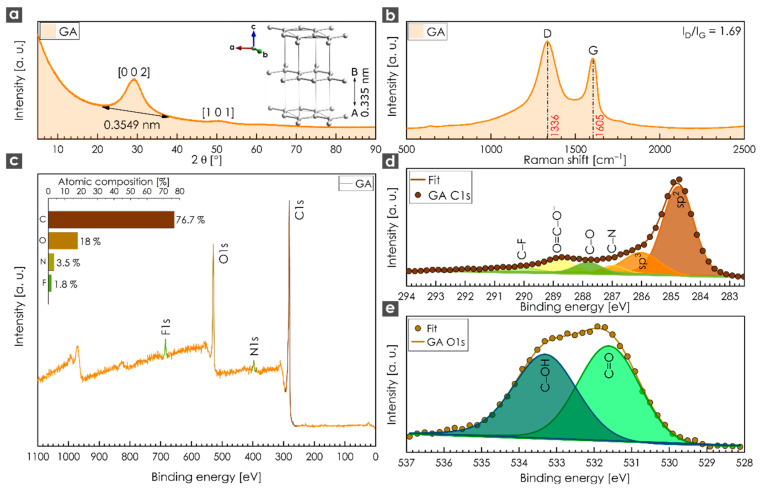
(**a**) X-ray diffraction (XRD) of graphene acid (GA); inset: crystal structure of graphite (hexagonal P 63 mc, *a*: 2.456 Å, *b*: 2.456 Å, *c*: 6.696 Å) obtained from the American Mineralogist Crystal Structure Database (database code: amcsd 0011247). The crystal structure was visualized with the program VESTA (version 3.4.7), developed by Koichi Momma and Fujio Izumi [50]. (**b**) Raman spectrum of GA with corresponding *I*_D_/*I*_G_ ratio; (**c**) X-ray photoelectron spectroscopy (XPS) survey spectrum and elemental composition (inset) of GA; (**d**) XPS spectra of C 1s and (**e**) O 1s derived from the GA sample.

**Figure 2 nanomaterials-10-01731-f002:**
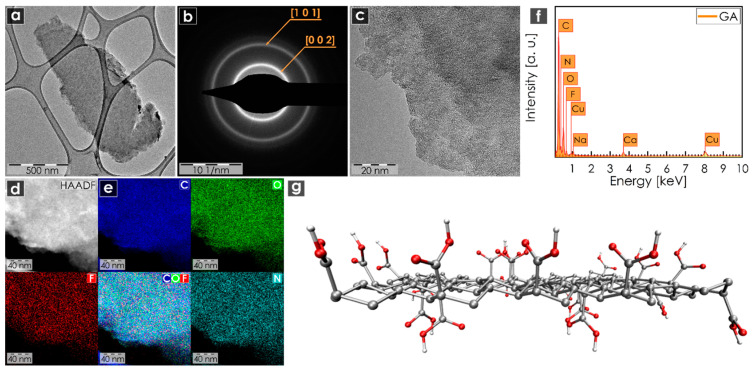
(**a**) High-resolution TEM (HRTEM) image of a GA sample; (**b**) selected area electron diffraction (SAED) analysis of a GA sample; (**c**) high-magnification HRTEM image of a GA sample; (**d**) dark-field HRTEM image used for EDS chemical mapping; (**e**) EDS elemental mapping of GA sample (on a Cu grid), including carbon, oxygen, fluorine and nitrogen atoms; (**f**) energy-dispersive X-ray (EDAX) spectrum of a GA sample; (**g**) optimized structure of GA. The molecular structure of GA was visualized using VMD software (version 1.9.3) [51]. Scale bars were unified from the graphical point of view in order to possess better visibility.

**Figure 3 nanomaterials-10-01731-f003:**
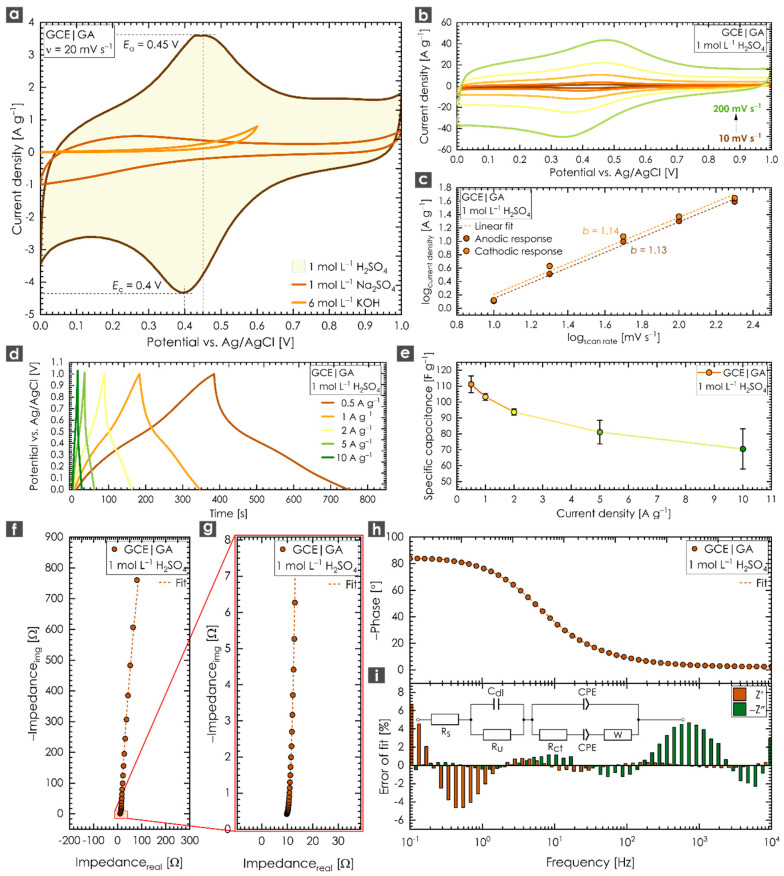
(**a**) Cyclic voltammetry (CV) response of a glassy carbon electrode (GCE) modified with GA in different electrolytes recorded at a constant scan rate of 20 mV s^−1^; (**b**) CVs for a GCE modified with GA at different potential scan rates ranging from 10 to 200 mV s^−1^; (**c**) calculation of *b* factor from a linear fit when log *i* is plotted against log *v*; (**d**) galvanostatic charging/discharging (GCD) response of a GCE modified with GA recorded at different current densities ranging from 0.5 to 10 A g^−1^; (**e**) specific capacitance evolution with increasing current density (0.5 to 10 A g^−1^) for three independent syntheses of GA; (**f**) Nyquist plot of a GCE modified with GA; (**g**) magnified high-frequency region of the Nyquist plot for a GCE modified with GA; (**h**) Bode representation of a GCE modified with GA; (**i**) the error of fit for the equivalent circuit used; inset: a modified Frumkin–Melik–Gaykazyan circuit. All measurements were performed in 1 mol L^−1^ H_2_SO_4_ unless otherwise stated.

**Figure 4 nanomaterials-10-01731-f004:**
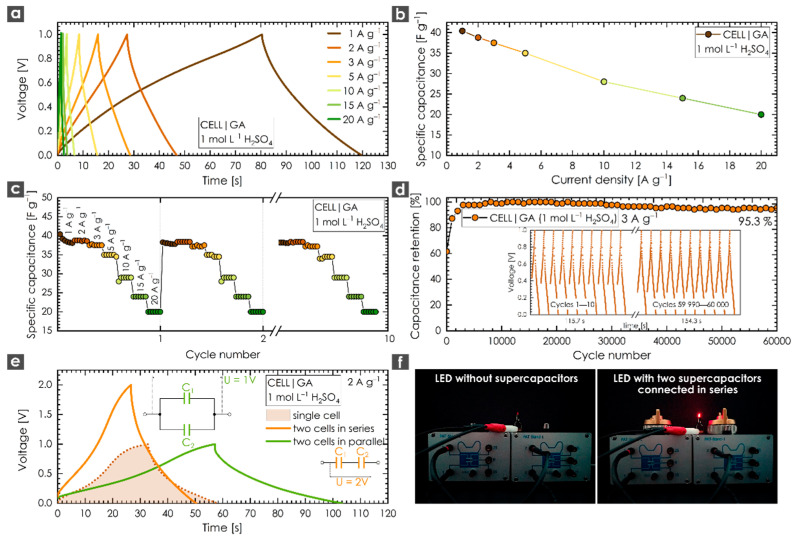
(**a**) GCD response of GA in a two-electrode configuration recorded at current densities ranging from 1 to 20 A g^−1^; (**b**) decrease in the specific capacitance of GA with increasing current density; (**c**) rate stability test of GA recorded at current densities ranging from 1 to 20 A g^−1^; (**d**) cycling stability of GA after 60,000 cycles; inset shows first and last 10 cycles in a potential difference of 1 V (recorded from −0.35 V to 0.65 V); (**e**) response of two supercapacitor cells equipped with GA connected in series or in parallel; (**f**) LED test of two supercapacitor cells equipped with GA connected in series. All measurements were performed in 1 mol L^−1^ H_2_SO_4_.

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
