# Peer review of "New Limits for Stability of Supercapacitor Electrode Material Based on Graphene Derivative"

_nanomaterials, 2020, doi:10.3390/nano10091731_

Round 1
Reviewer 1 Report
Please see the attached file

Reviewer 2 Report
There is a little contradiction at the beginning of the abstract where supercapacitors are said to feature excellent cycling stability and immediately this is possed under question.
The authors used the four-point probe method to test the conductivity of GA. For this method form factor and sample size are parameters to be considered carefully; I recommend developing this part.
Figure 4.e seems to be quite obvious as a capacitor device it is. What is the information that can be extracted from it?
Figure 4.f (left) shows "LED without supercapacitor", what the authors mean by that? Is it just an open circuit LED?
A quantitative comparison with other approaches is missing to evaluate the importance of the achievements.
Round 2
Reviewer 1 Report
The authors have revised their manuscript according to the referee’s comments. Therefore, the paper can be accepted in present form.